# Antimicrobial Peptide Octoprohibitin-Encapsulated Chitosan Nanoparticles Enhanced Antibacterial Activity against *Acinetobacter baumannii*

**DOI:** 10.3390/pharmaceutics16101245

**Published:** 2024-09-25

**Authors:** E. H. T. Thulshan Jayathilaka, Jinwook Han, Mahanama De Zoysa, Ilson Whang

**Affiliations:** 1College of Veterinary Medicine and Research Institute of Veterinary Medicine, Chungnam National University, Yuseong-gu, Daejeon 34134, Republic of Korea; thimira.thulshan@o.cnu.ac.kr; 2National Marine Biodiversity Institute of Korea (MABIK), 75, Jangsan-ro 101 beon-gil, Janghang-eup, Seocheon 33662, Republic of Korea; hiclow@mabik.re.kr

**Keywords:** *Acinetobacter baumannii*, Chitosan nanoparticles, encapsulation, Octoprohibitn-CNPs, drug delivery system

## Abstract

**Background:** This study focused on evaluating the physiochemical characteristics and antibacterial activity of Octoprohibitin-encapsulated CNPs (Octoprohibitin-CNPs) against *Acinetobacter baumannii*. **Methods:** Octoprohibitin was encapsulated into CNPs via ionotropic gelation with carboxymethyl chitosan (CMC) and low molecular weight chitosan (CS). Octoprohibitin-CNPs were dispersed in phosphate-buffered saline and the release kinetic profile was determined. Then Octoprohibitin-CNPs were examined using field-emission transmission electron microscopy and physicochemical characterization was performed. Antibacterial activity of Octoprohibitin-CNPs against *A. baumannii* was evaluated. Biofilm inhibition and eradication assays were performed using the crystal violet (CV) staining-based method for biofilm quantification. **Results**: The average diameter, zeta potential, encapsulation efficiency, and loading capacity of Octoprohibitin-CNPs were 244.5 ± 21.97 nm, +48.57 ± 0.38 mV, and 85.7% and 34.2%, respectively. TEM analysis imaging revealed that Octoprohibitin-CNPs are irregularly shaped, with fewer aggregates than CNPs. Octoprohibitin-CNPs exhibited a biphasic release pattern, characterized by an initial rapid phase followed by a sustained release over time, extending up to 93.68 ± 6.48% total release until 96 h. In vitro, Octoprohibitin-CNPs showed lower cytotoxicity compared to Octoprohibitin alone. Time-kill kinetic and bacterial viability reduction assays showed Octoprohibitin-CNPs exhibited slightly higher antibacterial activity against *A. baumannii* than Octoprohibitin. **Conclusions:** Octoprohibitin-CNP-treated *A. baumannii* exhibited higher levels of morphological deviation, increased membrane permeability, and the production of reactive oxygen species, as well as antibiofilm activity with greater biofilm inhibition and eradication than Octoprohibitin. These findings show that Octoprohibitin-CNPs perform better against *A. baumannii* compared to Octoprohibitin alone.

## 1. Introduction

*Acinetobacter baumannii* is a highly infectious bacterium that can lead to various complications, such as pneumonia, meningitis, septicemia, urinary tract infections, and abscesses [1]. Based on its pathogenicity, it is classified as a member of the ESKAPE group, which comprises *Enterococcus faecium*, *Staphylococcus aureus*, *Klebsiella pneumoniae*, *A. baumannii*, *Pseudomonas aeruginosa*, and *Enterobacter* spp., which are species known for their transmissibility [2]. Controlling *A. baumannii* has become increasingly challenging as it exhibits resistance to multiple antibiotics through various defense mechanisms, including the acquisition of β-lactamases, up-regulation of multidrug efflux pumps, modification of aminoglycosides, permeability defects, and alteration of target sites [3,4]. The consistent formation of biofilms by *A. baumannii* poses a significant challenge to effectively combating bacteria using existing antibiotics. These biofilms act as protective barriers that hinder the efficacy of antibiotics against bacteria [5]. In response, scientists are actively seeking alternative solutions to combat multidrug-resistant (MDR) *A. baumannii*. Recent studies have demonstrated that antimicrobial peptides (AMPs) are a promising option for controlling antibiotic-resistant bacteria [6].

AMPs are positively charged, short-chain amphipathic peptides (<50 amino acids) naturally produced by the host as defense molecules with diverse modes of action against various infections [7]. These include disrupting cell membranes, inhibiting cell wall synthesis, disrupting biofilms, modulating intracellular targets, inhibiting protein synthesis, binding to intracellular targets, and influencing host immunomodulation [8]. The diverse mechanisms of action of AMPs present a challenge for pathogen resistance, making them a favorable choice for addressing multidrug-resistant pathogens [9]. Nevertheless, the clinical application of AMPs has been constrained by various challenges, including their low stability and short half-life owing to degradation, toxicity, adverse reactions, and delivery limitations [10]. To overcome these challenges, several studies have focused on encapsulating AMPs into nanoparticles (NPs) using biocompatible materials [11].

Chitosan (CS), a linear polysaccharide derived from chitin, is composed of randomly distributed β-linked D-glucosamine and N-acetyl-D-glucosamine units [12]. Therefore, it holds significant promise as an NP-encapsulating agent in the field of biomedicine. CS has advantageous properties such as biodegradability, biocompatibility, and controlled release, making it a potential candidate for encapsulating AMPs [13]. Additionally, it exhibits inherent antimicrobial activity, offering potential synergistic benefits by enhancing the antimicrobial effects of AMPs through CS encapsulation [14]. Zhu et al. [11] used a similar approach and demonstrated extended and enhanced antimicrobial properties with low cytotoxicity when AMP (PFKLSLHL-NH2) was loaded into chitosan-based chitosan quaternary ammonium salt.

Octoprohibitin is a synthetic AMP derived from prohibitin-2 of *O. minor*, containing 26 AA with +10 net charge and a hydrophobic ratio of 38%. This study focused on developing an efficient drug delivery method to enhance the effectiveness of AMP while reducing toxicity and improving drug action. This was achieved by encapsulating Octoprohibitin in chitosan nanoparticles (CNPs) and evaluating its antimicrobial potency against *A. baumannii* [15]. The initial characterization of Octoprohibitin-loaded CNPs (Octoprohibitin-CNPs) involved the evaluation of the encapsulation efficiency percentage (EE%), loading capacity percentage (LC%), release profile of Octoprohibitin, and cytotoxicity. Subsequently, we compared the antimicrobial activity of Octoprohibitin-CNPs with that of Octoprohibitin using time-kill kinetics and viability assays. The intensity of the antimicrobial modes of action was assessed by examining bacterial morphology, membrane permeability, and reactive oxygen species (ROS) generation. Finally, the antibiofilm activity of Octoprohibitin-CNPs was determined and compared to that of Octoprohibitin.

## 2. Materials and Methods

### 2.1. Chemical Reagents

CS (Sigma-Aldrich, St. Louis, MO, USA) in 1% (*v*/*v*) acetic acid (pH 5), and a CMC (Kraeber & Co GmbH, Ellerbek, Germany) solution (1 mg/mL; pH 7.4) was prepared by dissolving CMC in distilled water. Octoprohibitin (1 mg/mL) was dissolved in nuclease-free water in sterile conditions. Additionally, the following chemicals were used. Uranyl acetate (Sigma-Aldrich, St. Louis, MO, USA), 3-(4,5-dimethylthiazol-2-yl)-2,5-diphenyltetrazolium bromide (MTT; Sigma-Aldrich, St. Louis, MO, USA). Propidium iodide (PI; Sigma-Aldrich, St. Louis, MO, USA). 2′7′dichlorodihydro-fluorescein diacetate (H_2_DCFDA; Sigma-Aldrich, St. Louis, MO, USA).

### 2.2. Optimization of the Octoprohibitin-CNPs Encapsulation and Determination of EE% and LC%

To derive the best EE% and LC% of Octoprohibitin-CNPs, a variable ratio of Octoprohibitin was used for ionotropic gelation optimization in the presence of CMC and CS at a constant level. According to the method described by Piras et al. [16], ionotropic gelation was conducted using Octoprohibitin: CMC: CS (Appendix A). The preparation of the solution, ionotropic gelation, and isolation of encapsulated nanoparticles were carried out following the method previously described [15]. The supernatant was collected to measure the remaining peptide and determine the EE% and LC%. The peptide concentration was measured using a Nanodrop (Thermo Fisher, Waltham, MA, USA), and the EE% and LC% were calculated using the formulas mentioned directly below. Sedimented Octoprohibitin-CNPs were resuspended in 1× phosphate-buffered saline (PBS). The NPs sizes were analyzed using a Mastersizer 3000 laser diffraction particle size analyzer (Malvern Panalytical, Malvern, UK).
EE% = (Initial weight of Octoprohibitin − Remained weight of Octoprohibitin in supernatant) × 100Initial weight of Octoprohibitin
LC% = (Initial weight of Octoprohibitin − Remained weight of Octoprohibitin in supernatant) × 100Total amount of CS and CMC used for encapsulation

### 2.3. Analysis of Octoprohibitin Release Kinetic Profile from Octoprohibitin-CNPs

The formulation with the highest EE% and optimal particle size was selected for further encapsulation of Octoprohibitin-CNPs and subsequent experiments. To evaluate the release kinetics, Octoprohibitin-CNPs containing 1 mg of Octoprohibitin were dispersed in 1 mL of solvent and the release kinetic profile was determined at 24 h intervals according to the method described previously [15]. The cumulative amount of released Octoprohibitin was calculated.

### 2.4. Morphological Analysis of Octoprohibitin-CNPs

Octoprohibitin-CNPs were examined using field-emission transmission electron microscopy (FE-TEM) to characterize their morphology. Both Octoprohibitin-CNPs and CNPs were suspended in PBS. Five microliters of each sample were placed on a formvar/carbon-coated copper grid and incubated for 10 min. The excess sample was removed by blotting with filter paper. Subsequently, five microliters of 2% uranyl acetate were applied to the grid for 5 s, and any remaining solution was removed by blotting. The grid was allowed to dry and then observed using FE–TEM (TecnaiTM G2 F30 super-twin (FEI), Hillsboro, OR, USA).

### 2.5. Cytotoxicity Analysis of Octoprohibitin-CNPs

The viability of human embryonic kidney 293T cells exposed to Octoprohibitin or Octoprohibitin-CNP was assessed using the MTT assay to compare the cytotoxicity. Cell culturing, treatment with Octoprohibitin and Octoprohibitin-CNPs (0–400 μg/mL), and the MTT assay were conducted according to the method described by Jayathilaka et al. [14].

### 2.6. Analysis of Antibacterial Activity of Octoprohibitin-CNPs against A. baumannii

Given the known antibacterial activity of Octoprohibitin against *A. baumannii*, a comparative analysis was conducted to evaluate the antibacterial efficacy of Octoprohibitin-CNPs versus Octoprohibitin. *A. baumannii* was cultured in tryptic soy agar or broth at 25 °C. Microdilution assay was performed to investigate the time-kill kinetics of *A. baumannii* according to the previously described method [14]. Octoprohibitin-CNPs and Octoprohibitin were treated at concentrations of 200 and 400 μg/mL. Apart from the peptide treatments, the CNPs (400 μg/mL) group was included, and the negative control (NC) and positive control (PC) were treated with PBS and 100 μg/mL of chloramphenicol, respectively, and incubated for 24 h. Bacterial growth was assessed by measuring OD_595_ every 3 h using a spectrophotometer.

The bacterial viability reduction efficacy between Octoprohibitin and Octoprohibitin-CNPs was compared using the MTT assay. Bacterial seeding and treatments were conducted similarly to the time-kill kinetic assay. After 24 h of incubation, the bacteria were collected by centrifugation at 1500× *g* for 10 min at 4 °C and washed with PBS. The MTT assay was conducted according to the method described in Section 2.4, and absorbance was measured at OD_595_ using a microplate spectrophotometer.

### 2.7. Analysis of Morphological Changes in A. baumannii Following Treatment with Octoprohibitin-CNPs

To assess the efficacy of Octoprohibitin-CNPs in inducing morphological changes in *A. baumannii*, field emission scanning electron microscopy (FE-SEM) was performed on bacteria treated with CNPs, Octoprohibitin, and Octoprohibitin-CNPs. Bacteria were cultured as described in Section 2.5 and treated with CNPs (400 µg/mL), Octoprohibitin (200 and 400 µg/mL), and Octoprohibitin-CNPs (200 and 400 µg/mL). After 12 h, FE-SEM was conducted according to the method described by Jayathilaka et al. [14].

### 2.8. Analysis of Membrane Permeability Alteration of A. baumannii with Octoprohibitin-CNPs Treatment

A propidium iodide uptake (PI) assay was conducted on *A. baumannii* treated with CNPs, Octoprohibitin, and Octoprohibitin-CNP to compare membrane permeability alterations. The bacteria were initially cultured and treated as described in Section 2.6, followed by PI staining. Confocal microscopy analysis was then conducted according to the method described previously [14].

### 2.9. Analysis of ROS Generation in A. baumannii Treated with Octoprohibitin-CNPs

To assess and compare ROS generation in *A. baumannii* treated with CNPs, Octoprohibitin, and Octoprohibitin-CNPs, the treated bacteria were subjected to staining with H_2_DCFDA. Staining and Confocal microscopy analysis were conducted according to the method described previously [14].

### 2.10. Analysis of Biofilm Inhibition and Eradication Activity of Octoprohibitin-CNPs in A. baumannii

Biofilm inhibition and eradication assays were performed using the crystal violet (CV) staining-based method for biofilm quantification. Octoprohibitin, and Octoprohibitin-CNPs at concentrations of 200 and 400 µg/mL were treated, and biofilm inhibition and eradication assays were conducted according to the method described previously [15]. Biofilm inhibition or eradication was quantified using the following formula:Biofilm inhibition/eradication = [1 − (Ab of the sample/Ab of the NC)] × 100%

### 2.11. Statistical Analysis

Data analyses were conducted using GraphPad Prism (version 8) for Windows and Mac (GraphPad Software Inc., San Diego, CA, USA). qRT-PCR results were analyzed using one-way analysis of variance and/or an unpaired *t*-test. Data are represented as the mean ± standard error (SE) of the mean of triplicate experiments.

## 3. Results

### 3.1. Optimization and Preparation of the Octoprohibitin-CNPs

The ionotropic gelation method was applied to encapsulate Octoprohibitin within a chitosan matrix. The optimal encapsulation ratio was determined by varying the Octoprohibitin concentration while maintaining a constant CS: CMC ratio. The EE% and LC% were calculated by quantifying the residual Octoprohibitin in the supernatant following NP isolation. Reaction mixture 4 (CS: CMC: Octoprohibitin-0.4:2:1) exhibited the highest EE (85.7%) (Appendix A). Although reaction mixture 5 (CS: CMC: Octoprohibitin, 0.4:2:1.5) demonstrated a higher LC (47.26%) compared to reaction mixture 3 (34.20%), it showed a lower EE (75.56%). Based on the EE%, a CS: CMC: Octoprohibitin ratio of 0.4:2:1 was identified as the optimal ratio for subsequent encapsulation experiments.

### 3.2. Characterization and Release Kinetics of Octoprohibitin-CNPs

A suspension of Octoprohibitin-CNPs was formed, containing small particle aggregates. After brief sonication, the Octoprohibitin-CNPs were completely dispersed, with no visible aggregates. The laser diffraction particle size analysis revealed that the diameter of CNPs and Octoprohibitin-CNPs was 444.5 ± 21.97 and 246.8 ± 1.98 nm, respectively. The CNPs and Octoprohibitin-CNPs exhibited a zetapotential of +59.33 ± 3.63 mV and +48.57 ± 0.38 mV, respectively, confirming the cationic nature of the NPs in PBS at pH 7.4.

The release profile of Octoprohibitin from Octoprohibitin-CNPs was assessed in PBS at pH 7.4 over a period of 96 h (Figure 1A). The peptide release kinetics displayed a biphasic pattern, with an initial rapid release of 66.31% within the first 36 h, followed by a sustained release phase that reached a maximum of 93.68% after 96 h. FE-TEM analysis revealed the morphology of both CNPs and Octoprohibitin-CNPs (Figure 1B). TEM micrographs of the CNPs revealed round, aggregated particles, whereas Octoprohibitin-CNPs exhibited irregularly shaped, non-aggregated NPs.

### 3.3. Cytotoxicity of Octoprohibitin-CNPs and Octoprohibitin

Assessment of the cytotoxicity in HEK cells treated with Octoprohibitin-CNPs or Octoprohibitin revealed no notable reduction in viability up to 25 μg/mL for both treatments (Figure 2). However, Octoprohibitin at concentrations ranging from 50 to 400 μg/mL exhibited a significant decrease (*p* < 0.05) in viability compared to Octoprohibitin-CNPs, with the lowest viability of 83.14% observed at 400 μg/mL. In contrast, the viability of cells treated with Octoprohibitin-CNPs remained unchanged up to 400 μg/mL.

### 3.4. Antibacterial Activity of Octoprohibitin versus Octoprohibitin-CNPs

In our previous study, we reported the antibacterial activity of Octoprohibitin against *A. baumannii*, with a minimum inhibitory concentration (MIC) of 200 μg/mL and a minimum bactericidal concentration (MBC) of 400 μg/mL. To evaluate the antibacterial activity of Octoprohibitin-CNPs, time-kill kinetics and bacterial viability reduction assays were conducted with both Octoprohibitin and Octoprohibitin-CNPs. In the time-kill kinetic assay, CNPs demonstrated the lowest antibacterial activity compared to the NC, with a maximum OD_595_ of 0.84 (Figure 3A). At the MIC level (200 μg/mL), Octoprohibitin-CNPs exhibited slightly higher bacterial growth inhibition (OD_595_; 0.10) compared to Octoprohibitin (OD_595_; 0.19). Nevertheless, at the MBC level, Octoprohibitin and Octoprohibitin-CNPs exhibited comparable degrees of bacterial growth inhibition, with OD_595_ ranging from 0.63 to 0.75. Interestingly, in the bacterial viability reduction assay, Octoprohibitin-CNPs demonstrated a slightly higher viability reduction at both the MIC and MBC levels, with percentages of 39.02% and 20.01%, respectively (Figure 3B). In contrast, Octoprohibitin caused viability reductions of 43.84% and 24.62% at the MIC and MBC, respectively, which shows relatively lower inhibitory effects.

### 3.5. Morphological Alterations of A. baumannii Treated with Octoprohibitin and Octoprohibitin-CNPs

*A. baumannii* treated with Octoprohibitin and Octoprohibitin-CNPs were examined using FE-SEM to assess the extent of morphological changes. The NC group displayed the typical morphology of *A. baumannii*, while treatment with CNPs caused slight alterations, including surface shrinkage (Figure 4). At the MIC level, both Octoprohibitin and Octoprohibitin-CNPs induced similar morphological alteration, characterized by slight shrinkage and rough surfaces (Figure 4B,E). However, at the MBC level, the Octoprohibitin-CNP-treated sample displayed more pronounced pore formation and total cell disruption compared to the Octoprohibitin-treated samples (Figure 4C,D).

### 3.6. Alteration of Membrane Permeability in A. baumannii Treated with Octoprohibitin-CNPs Compared to Octoprohibitin

PI uptake assay was conducted to evaluate the membrane permeability of *A. baumannii* following treatment with Octoprohibitin and Octoprohibitin-CNPs. Viable cells stained with FDA emitted green fluorescence, while cells with altered membrane permeability exhibited red fluorescence when stained with PI. In both the NC- and CNP-treated cells, the majority displayed green fluorescence, indicating no significant alterations in membrane permeability (Figure 5). At the MIC, both Octoprohibitin and Octoprohibitin-CNPs exhibited low levels of red fluorescence and high levels of green fluorescence, confirming moderate alterations in membrane permeability for both treatments at this concentration. At the MBC level, both Octoprohibitin and Octoprohibitin-CNPs displayed high levels of red fluorescence and no green fluorescence, confirming membrane permeability alterations across the entire bacterial population.

### 3.7. ROS Generation in A. baumannii with Octoprohibitin-CNPs Compared to Octoprohibitin

The ROS generation in *A. baumannii* treated with Octoprohibitin-CNPs and Octoprohibitin was assessed using H_2_DCFDA staining. The appearance of green fluorescence in bacterial cells indicated induced ROS production. Bacteria treated with CNPs displayed no green fluorescence, similar to the negative control, confirming that CNPs alone did not induce ROS generation (Figure 6). However, at the MIC, Octoprohibitin-CNPs exhibited a slightly higher level of green fluorescence compared to Octoprohibitin alone. The highest level of green fluorescence was observed at the MBC for Octoprohibitin-CNPs, closely followed by the MBC for Octoprohibitin, confirming that Octoprohibitin-CNPs have superior ROS generation capability over Octoprohibitin.

### 3.8. Antibiofilm Activity of Octoprohibitin-CNPs Compared to Octoprohibitin in A. baumannii

A biofilm inhibition assay was performed to assess the inhibitory effects of Octoprohibitin-CNPs and Octoprohibitin on *A. baumannii* biofilm formation. Notably, CNPs alone demonstrated biofilm inhibition at concentrations of 100 and 200 μg/mL, with inhibition rates of 24.84% and 34.32%, respectively (Figure 7). At these concentrations, Octoprohibitin-CNPs showed greater inhibition levels compared to Octoprohibitin, achieving 59.25% and 41.78% inhibition at the MIC level and 84.53% and 75.79% inhibition at the MBC level, respectively.

A biofilm eradication assay was conducted to evaluate the disruptive effects of Octoprohibitin-CNPs and Octoprohibitin on pre-formed biofilm of *A. baumannii*. CNPs exhibited minimal biofilm eradication activity at both 100 and 200 μg/mL, achieving approximately 3.60% inhibition. In contrast, Octoprohibitin-CNPs and Octoprohibitin demonstrated significant concentration-dependent biofilm eradication. The highest biofilm eradication was achieved with Octoprohibitin-CNPs at 200 μg/mL, resulting in 87.30% eradication, while Octoprohibitin at the same concentration resulted in 80.53% eradication.

## 4. Discussion

Conjugates of AMPs with biopolymers can self-assemble into various structures, including NPs, hydrogels, micelles, and vesicles. These structures not only preserve exceptional antimicrobial activities but also support the mitigation of toxicity [11]. In this study, an ionotropic gelation method was applied to encapsulate Octoprohibitin into CNPs, to improve their stability and antimicrobial efficacy while mitigating their toxic characteristics. Encapsulating positively charged Octoprohibitin in positively charged CS presents a significant challenge due to electrical repulsion and restricted encapsulation efficiency. To address this issue, we introduced anionic CMC as a crosslinker to facilitate the encapsulation of Octoprohibitin and maintain the integrity of the CNPs. Yuning et al. [17] demonstrated the dual encapsulation of AMPs in a topical delivery system using dendritic nanogels (DNGs), followed by their incorporation into a poloxamer gel. Similarly, during the initial stirring process, Octoprohibitin combined with CMC formed macroaggregates of Octoprohibitin-encapsulated CMC particles. The subsequent introduction of CS into the medium facilitated the formation of Octoprohibitin-CNPs.

Recently, various biomaterials have been used to encapsulate AMPs. For example, synthetic AMPs, namely GIBIM-P5S9K and GAM019, encapsulated within poly lactic-co-glycolic acid (PLGA) NPs exhibited characteristics such as a mean diameter (290 nm), a negative zeta potential, and a remarkable EE% (>90%) [18]. Furthermore, chitosan-encapsulated cecropin-A (1–7)-melittin-cell-penetrating peptide had a diameter of 597.5 nm, positive zeta potential (2.8 mV), and EE% of 75.1% [19]. In comparison to these studies, the encapsulation process of Octoprohibitin-CNPs achieved a high EE% of 85.7% and a substantial LC of 34.2%, confirming successful encapsulation. Furthermore, Octoprohibitin-CNPs had a mean diameter of 444.5 nm, and a zeta potential of 59.3 mV. Although both were in the nanoscale range, Octoprohibitin-CNPs had a larger particle size and a lower zeta potential than CNPs. The positive charge (higher zeta potential) indicates a stronger attraction of the Octoprohibitin-CNPs toward the negatively charged bacteria membrane, facilitating enhanced Octoprohibitin delivery. In addition, in the FE-TEM analysis, Octoprohibitin-CNPs showed fewer aggregations than CNPs, suggesting that encapsulation might improve the dispersion of the final product. These differences can be attributed to electrostatic charge rearrangement and conformational changes in the NPs during ionotropic gelation. Similarly, Somaye et al. [20] demonstrated that LL37-encapsulated CNPs exhibit increased particle size, decreased aggregation level, and reduced zeta potential compared to CNPs alone. A biphasic release profile of CS-based NPs, characterized by an initial rapid release followed by a sustained release of the encapsulated drugs, is crucial for prompt drug action and for maintaining drug concentration over an extended period, ensuring both immediate and prolonged drug efficacy [21]. In our study, Octoprohibitin-CNPs exhibited a release profile featuring an initial rapid release of Octoprohibitin up to 48 h, followed by a sustained release that continued until 96 h. This design allows for a rapid increase in the therapeutic concentration at the targeted site while maintaining the concentration through a slow-release mechanism throughout the treatment period.

Although Octoprohibitin did not induce significant toxicity within the therapeutic dose range, in vitro studies revealed concentration-dependent reductions in cellular viability, with an 83% reduction at 400 μg/mL in HEK293 cells. Additionally, hemolytic activity was observed at concentrations of 800 μg/mL and above [15]. One of the main objectives of this study was to mitigate Octoprohibitin toxicity by encapsulating it in CNPs, since the sustained release profile of encapsulated NPs helps to reduce AMP-associated toxicity [22]. Ron et al. [23] demonstrated that LL-37 encapsulated in PEGylated liposomes enhanced bioavailability and ~19-fold lower cytotoxicity against keratinocytes compared to LL-37 alone. Similarly, we observed significant (*p* < 0.05) cytotoxicity reduction of Octoprohibitin-CNPs compared to Octoprohibitin above 50 μg/mL level. These findings suggest the potential to administer higher Octoprohibitin treatment dosages using Octoprohibitin-CNPs while maintaining a favorable safety profile.

In the medical field, CS exhibits notable bioactive effects, including antioxidant, anti-inflammatory, and wound-healing properties, as well as inherent antimicrobial characteristics [12]. Moreover, CS exhibits synergistic antimicrobial activity when used as an encapsulation agent for AMPs. AMP dendrimer-chitosan polymer conjugates, including NPs, gels, and foams, exhibit a synergistic effect against *Pseudomonas aeruginosa* by damaging the outer and inner Gram-negative bacterial membranes, with the added benefit of no toxicity [24]. In this study, Octoprohibitin-CNPs exhibited enhanced antibacterial activity against *A. baumannii* in time-kill kinetics and bacterial viability assays compared to Octoprohibitin alone. This finding confirms that Octoprohibitin-CNPs exhibit synergistic antibacterial action. In a previous study, we demonstrated the diverse modes of action of Octoprohibitin in inducing antibacterial activity, including morphological changes, alterations in membrane permeability, and ROS generation. In the current study, we observed that these modes of action were enhanced when treating *A. baumannii* with Octoprohibitin-CNPs. Positively charged CNPs electrostatically attract lipopolysaccharide-rich, negatively charged Gram-negative bacteria and coat their surfaces with NPs [25]. Additionally, the nanoscale size facilitates easy penetration of the bacterial wall and membrane, allowing the cargo to be internalized into bacterial cells [13]. Moreover, CS treatment induces osmotic pressure changes in bacteria, leading to membrane disruption and shrinkage [26]. These findings elucidate the high affinity of Octoprohibitin-CNPs for *A. baumannii*, resulting in synergetic damage to the bacterial membrane and increased membrane permeability. FE-SEM analysis confirmed these findings by revealing a higher degree of visible pore formation and shrinkage in bacteria treated with Octoprohibitin-CNPs compared to those treated with Octoprohibitin alone. CS is recognized as an effective chelating agent for various biomolecules, including metal ions, which act as catalysts for ROS generation [27]. Our previous study illustrated that Octoprohibitin can induce ROS production in *A. baumannii*, which facilitates oxidative stress in bacteria and, ultimately, bactericidal activity [15]. In the present experimental setup, encapsulation with CS significantly enhanced the internalization of Octoprohibitin, resulting in increased ROS generation and greater bacterial death due to oxidative stress.

A pivotal virulence factor of *A. baumannii* is its ability to form biofilms, which contribute to the bacteria’s resistance to antimicrobial agents and enhance their persistence [5]. The polycationic properties of the amino groups allow CS to bind to planktonic bacteria. This binding enables CS to interact with negatively charged components essential for biofilm formation, including extracellular polymeric substances, proteins, and DNA, thereby inhibiting bacterial biofilm formation [28]. Moreover, the chelation of metal ions such as calcium, zinc, and magnesium disrupt gene expression, leading to bacterial death before biofilm formation [28,29]. Additionally, the presence of CS in encapsulated CNPs enhances the penetration of AMP into the biofilm matrix, thereby promoting the effective eradication of the biofilm [29]. For instance, the CS-streptomycin conjugate/gold NPs exhibited significant antibiofilm potency against Gram-negative bacteria such as *P. aeruginosa* and *Salmonella enterica*, as well as Gram-positive bacteria including *Listeria monocytogenes* and *Staphylococcus aureus* [30]. Similarly, in the present study, Octoprohibitin-CNPs demonstrated a greater inhibition and eradication of biofilm formation compared to Octoprohibitin alone, confirming that encapsulation in CS enhanced the antibiofilm activity of Octoprohibitin.

In summary, encapsulating Octoprohibitin into CNPs proved to be a successful strategy for delivering Octoprohibitin to both the planktonic and biofilm stages of multidrug-resistant *A. baumannii*, effectively controlling the bacteria. The biphasic release profile of Octoprohibitin-CNPs enabled both rapid and sustained bacterial control. The incorporation of both CS and Octoprohibitin in Octoprohibitin-CNPs offered additional benefits, including enhanced antibacterial efficacy and reduced toxicity. This synergy resulted in morphological changes, alterations in membrane permeability, ROS generation, and enhanced antibiofilm activity in *A. baumannii*, surpassing the effects of Octoprohibitin alone. Future in vivo studies will validate the effectiveness of Octoprohibitin-CNPs. With further optimization and development of mass production methods, Octoprohibitin-CNPs will demonstrate the advantages of using Octoprohibitin as the final dosage form in therapeutic applications. These findings have paved the way for new approaches in targeted drug delivery using CS-NPs, not just for AMPs but also for other drugs and chemicals. This strategy enhances antimicrobial effectiveness while reducing toxicity and prolonging drug action.

## Figures and Tables

**Figure 1 pharmaceutics-16-01245-f001:**
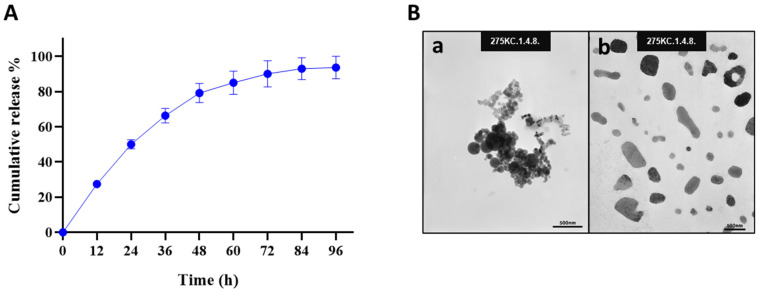
The cumulative release profile and morphology of Octoprohibitin-CNPs. (**A**) The cumulative release profile of Octoprohibitin from Octoprohibitin-CNPs in PBS (pH 7.4) at 37 °C presented as mean ± SD (n = 3). (**B**) Ultrastructural morphology of CNPs and Octoprohibitin-CNPs as observed under the field emission transmission electron microscope (FE-TEM). Micrographs show (**a**) CNPs and (**b**) Octoprohibitin-CNPs.

**Figure 2 pharmaceutics-16-01245-f002:**
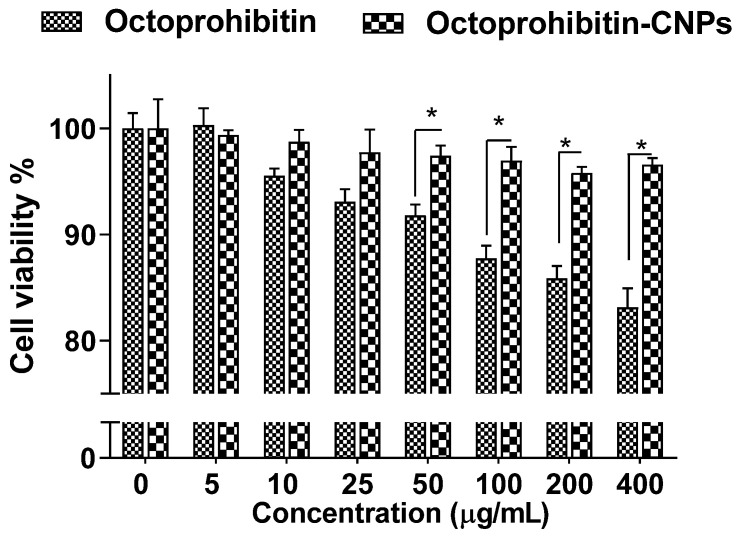
Cytotoxicity of Octoprohibitin versus Octoprohibitin-CNPs in human embryonic 293 cells as determined by MTT assay. Cells at 2.0 × 10^5^ cells/mL were seeded into a 96-well plate and incubated for 12 h. Cells were then treated with various concentrations of Octoprohibitin and Octoprohibitin-CNPs (0–400 µg/mL) and incubated at 37 °C for 24 h in a 5% CO_2_ incubator. The MTT assay was used to determine cell viability. * *p* < 0.05. Error bars represent the mean ± standard deviation (n = 3).

**Figure 3 pharmaceutics-16-01245-f003:**
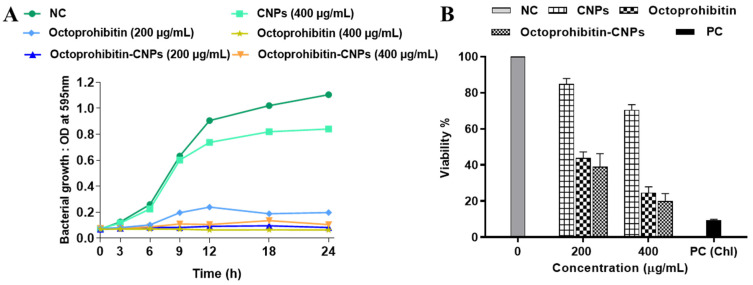
Effect of Octoprohibitin-CNPs in bacterial growth kinetics and cell viability against *A. baumannii*. CNPs, Octoprohibitin, and Octoprohibitin-CNPs were tested against *A. baumannii* at concentrations of 200 and 400 µg/mL. (**A**) Bacterial growth was determined by measuring the optical density at 595 nm at 0, 3, 6, 9, 12, 18, 24 h and time-kill kinetic curves were derived. (**B**) Bacterial viability was assessed using MTT assay after 24 h of treatment (mean ± SD, n = 3). Chloramphenicol (Chl) was used as a positive control (PC) for *A. baumannii*.

**Figure 4 pharmaceutics-16-01245-f004:**
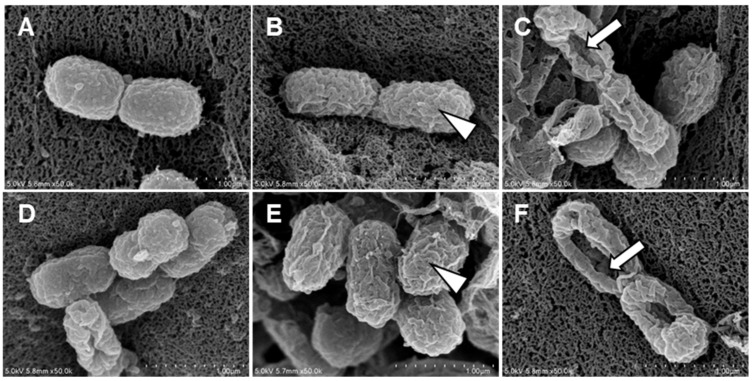
Comparison of morphological and ultrastructural alterations of *A. baumannii* following treatments with CNPs, Octoprohibitin, and Octoprohibitin-CNPs. *A. baumannii* was treated with (**A**) PBS, (**B**) Octoprohibitin (200 µg/mL), (**C**) Octoprohibitin (400 µg/mL), (**D**) CNPs (400 µg/mL), (**E**) Octoprohibitin-CNPs (200 µg/mL) (**F**) Octoprohibitin-CNPs (400 µg/mL). After 12 h of treatment, the bacteria were fixed with glutaraldehyde, coated with platinum, and then observed under FE-SEM. Scale bar 1 µm. Arrowheads indicate cell surface shrinkage, while arrows highlight pore formation on the bacteria.

**Figure 5 pharmaceutics-16-01245-f005:**
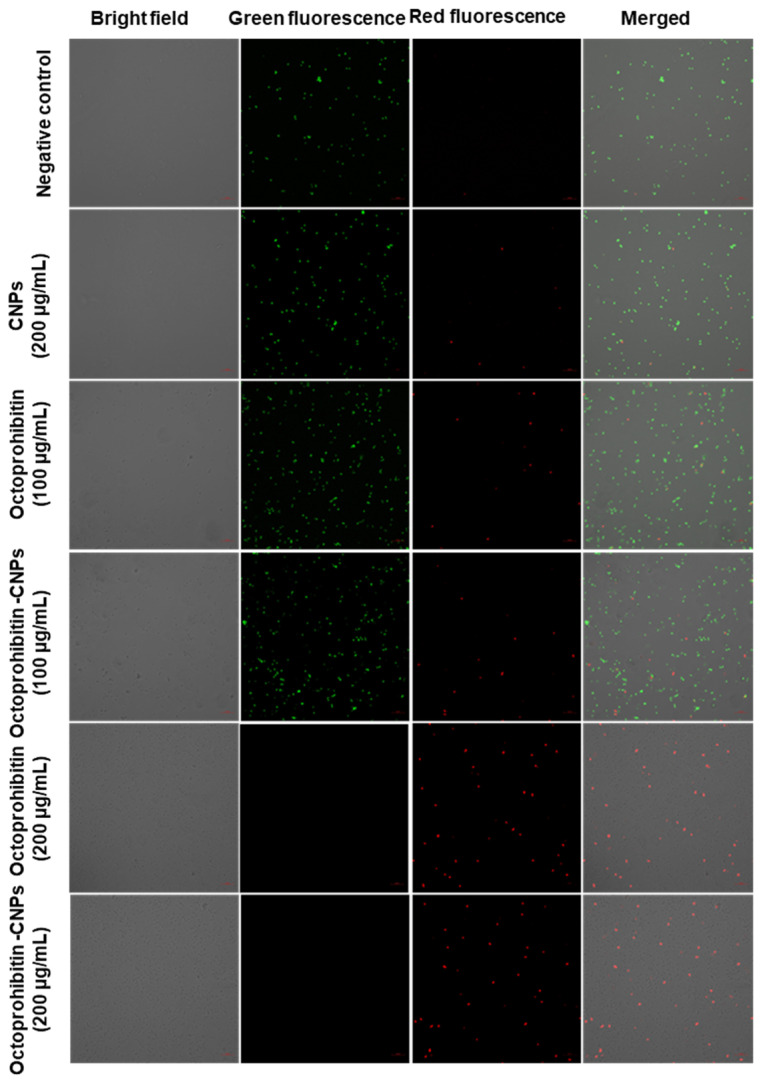
Comparison of membrane permeability alterations in *A. baumannii* with treatments of CNPs, Octoprohibitin, and Octoprohibitin-CNPs. *A. baumannii* was treated with PBS (negative control), CNPs, Octoprohibitin (200 and 400 µg/mL), and Octoprohibitin-CNPs (200 and 400 µg/mL). After 12 h of treatment, bacteria were separated and stained with PI to assess membrane permeability and FDA to determine cell viability. Bacteria were observed under confocal microscopy at excitation and emission wavelengths of 488 and 535 nm for green fluorescence and 535 and 617 nm for red fluorescence, respectively.

**Figure 6 pharmaceutics-16-01245-f006:**
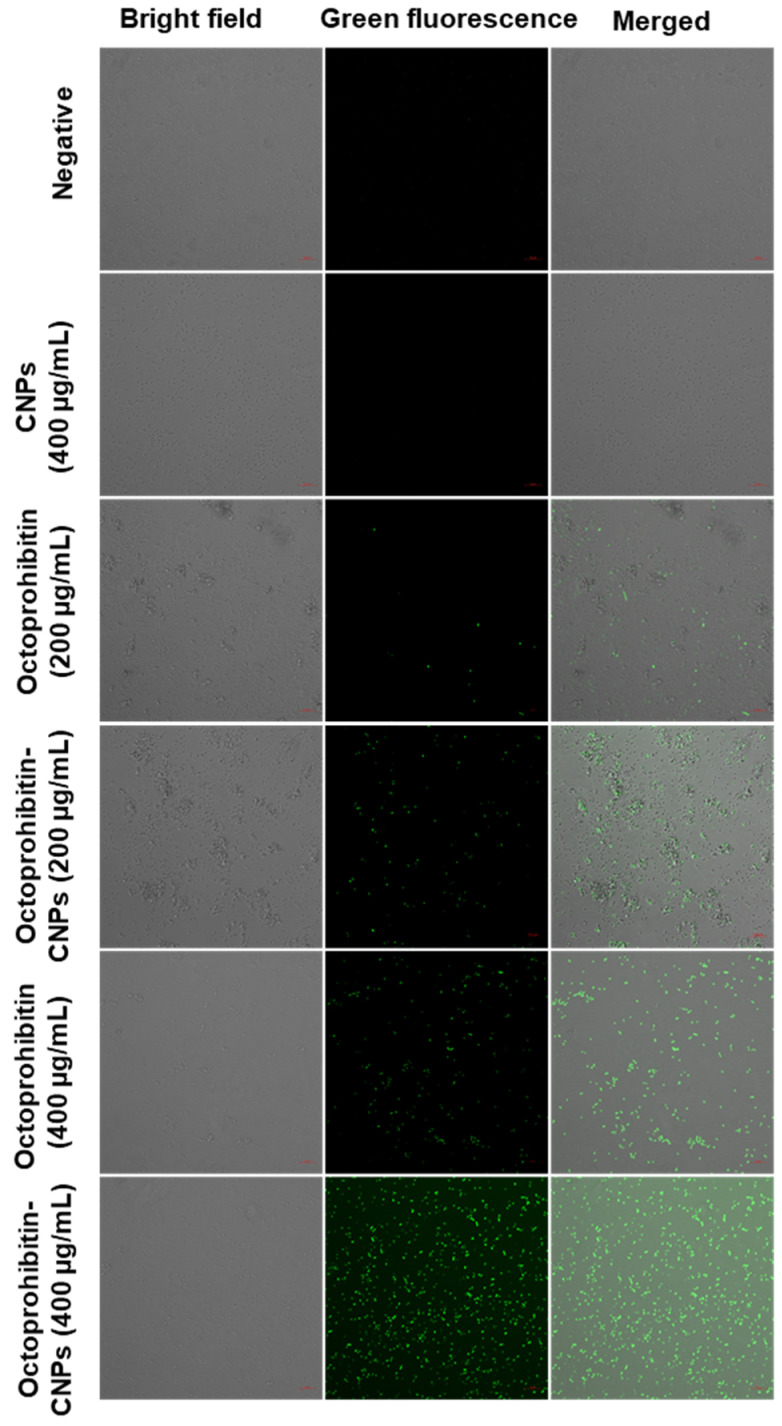
Comparison of reactive oxygen species (ROS) generation in *A. baumannii* treated with CNPs, Octoprohibitin, and Octoprohibitin-CNPs. *A. baumannii* was treated with PBS (negative control), CNPs (400 µg/mL), Octoprohibitin (100 and 200 µg/mL), and Octoprohibitin-CNPs. After 12 h of treatment, bacteria were separated and stained with 2′,7′-dichlorodihydrofluorescein diacetate (H2DCFDA). Bacteria were then observed under confocal microscopy at excitation and emission wavelengths of 535 and 617 nm, respectively.

**Figure 7 pharmaceutics-16-01245-f007:**
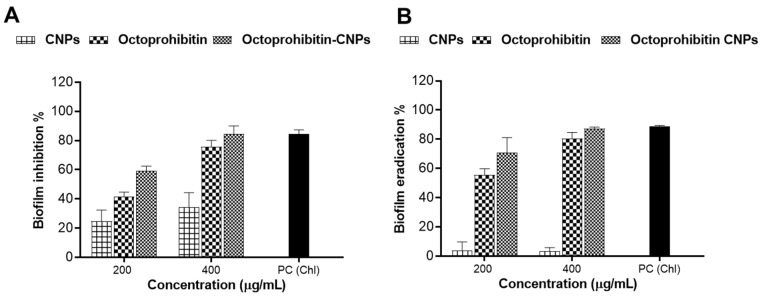
Comparison of the antibiofilm activity of Octoprohibitin-CNPs in *A. baumannii* against CNPs and Octoprohibitin. (**A**) *A. baumannii* culture was treated with CNPs, Octoprohibitin, and Octoprohibitin-CNPs, and the biofilm inhibition levels for each treatment were quantified using the crystal violet staining method. (**B**) A biofilm eradication assay was performed on *A. baumannii* biofilms treated with CNPs, Octoprohibitin, and Octoprohibitin-CNPs. Biofilm eradication was quantified via crystal violet staining.

## Data Availability

Data is contained within the article or Appendix A.

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
