# Peer review of "Antimicrobial Peptide Octoprohibitin-Encapsulated Chitosan Nanoparticles Enhanced Antibacterial Activity against Acinetobacter baumannii"

_pharmaceutics, 2024, doi:10.3390/pharmaceutics16101245_

Round 1

Reviewer 1 Report

Comments and Suggestions for Authors

In the current work, the authors have developed chitosan nanoparticles loaded with antimicrobial peptide Octoprohibitin and evaluated their safety and efficacy in in-vitro models. The chitosan nanoparticles were comprehensively characterized and needs further evaluations as under:

1.       Chitosan nanoparticles are made of semisynthetic polymer chitosan which shows batch to batch variability in its purity and has not received FDA’s generally recognized as safe (GRAS) status.  Could you please explain the rational behind choosing chitosan nanoparticles as opposed to lipid nanoparticles in delivering Octoprohibitin. 

2.       Octoprohibitin is water soluble molecule.  What was the rationale in formulating this molecule into a nanoparticle delivery system?

3.       Could you comment on how will you sterile the chitosan nanoparticles before administration?

Please include your responses to the above questions for further consideration.

Thanks for your valuable contribution.

Author Response

Please find the attached author's detailed responses. 

Reviewer 2 Report

Comments and Suggestions for Authors

Acinetobacter baumannii is a highly infectious and opportunistic bacterium known for causing severe infections, particularly in hospital settings. It is associated with a range of complications such as pneumonia, meningitis, septicemia, urinary tract infections, and wound infections. The paper explores how encapsulating the antimicrobial peptide Octoprohibitin in chitosan nanoparticles improves its effectiveness against the resistant bacterium Acinetobacter baumannii.

Although the methods of eradicating A. baumannii are interesting and up-to-date, the purpose of the study is not clearly stated. Please clarify the study's goal.

Detailed questions

1. The materials should be described in a separate section. 

2. Please justify the drug dissolution conditions.

3. Was the stability of CNPs determined?

4. Please discuss the consequences of zeta potential on the properties of CNPs.

5. What are the potential benefits of using chitosan as an encapsulating agent for AMPs?

6. Summarize the key findings of the study regarding the effectiveness of Octoprohibitin-CNPs against A. baumannii.

7. What implications do these findings have for the development of new antimicrobial treatments?

8. What future research directions are suggested by the authors to further explore the potential of Octoprohibitin-CNPs?

Author Response

(The authors gave the same response as above.)

Round 2

Reviewer 1 Report

Comments and Suggestions for Authors

Thank you for your responses to the requested comments.  They have been satisfactorily addressed.

Author Response

We have corrected sections as requested.